# Novel Approaches to the Design of an Ultra-Fast Magnetorheological Valve for Semi-Active Control

**DOI:** 10.3390/ma14102500

**Published:** 2021-05-12

**Authors:** Zbyněk Strecker, Filip Jeniš, Michal Kubík, Ondřej Macháček, Seung-Bok Choi

**Affiliations:** 1Institute of Machine and Industrial Design, Faculty of Mechanical Engineering, Brno University of Technology, Technicka 2, 616 69 Brno, Czech Republic; Filip.Jenis@vutbr.cz (F.J.); Michal.Kubik@vutbr.cz (M.K.); Ondrej.Machacek@vutbr.cz (O.M.); 2Department of Mechanical Engineering, The State University of New York at Korea (SUNY Korea), Incheon 21985, Korea

**Keywords:** magnetorheological valve, response time, eddy currents, magnetic simulations, SMC material

## Abstract

This article presents a list of suitable techniques and materials leading to the design of an ultra-fast magnetorheological (MR) valve. Two approaches for achieving the short response time are proposed: (a) by means of material, and (b) by means of the shape. Within the shape approach, the revolutionary technique of 3D metal printing using a selective laser melting (SLM) method was tested. The suitability of the materials and techniques is addressed based on the length of the response time, which is determined by the FEM. The simulation results determine the response time of the magnetic flux density on the step signal of the current. Subsequently, the response time is verified by the measurement of the simple magnetorheological valve. The following materials were tested: martensitic stainless steel AISI 420A (X20Cr13), cutting steel 11SMn30, pure iron for SLM, Sintex SMC STX prototyping material, ferrite N87, and Vacoflux 50. A special technique involving grooves was used for preventing eddy currents on materials with a high electrical conductivity. The simulation and experimental results indicate that a response time shorter than 2.5 ms can be achieved using materials such as Sintex SMC prototyping, ferrite N87, and grooved variants of metal pistons.

## 1. Introduction

In 1948, Jacob Rabinow discovered a new class of smart materials, currently known as magnetorheological fluids (MRFs) [1]. MRFs are generally made of three basic compounds: micron-sized iron particles, carrier oil, and additives. Immediately upon the application of an external magnetic field, the state of the MRF changes from a fluid to semi-solid or plastic. With the new state, the MRF exhibits a viscoplastic behavior, characterized by the initial stress (yield stress) that varies based on the magnitude of the applied magnetic field [2,3,4,5,6,7].

MRF is most commonly used in dampers, especially in (a) the primary suspension systems of vehicles [8,9,10,11,12,13], (b) the secondary suspension of driver seats in trucks [14,15,16,17,18], (c) the damping of vibration for seismic activity protection [19,20,21,22], or (d) the damping of cable bridge vibrations caused by wind and rain [23,24]. Semi-active control can be easily applied to magnetorheological (MR) dampers. Many research teams simulated semi-active algorithms only on the virtual quarter models or full models of the cars [25,26,27]. These simulations are working with different phenomenological models of MR dampers. One of the most complex models including the hysteretic behavior of an MR damper is described in paper [28]. The performance of semi-active algorithms depends on a dynamic force range, which should be as large as possible. There are papers [29,30,31] dealing with the optimal design of an MR valve using finite element analysis (FEA), but they are focused only on maximizing the static magnetic field range in the gap of the damper, and the dynamic behavior (response time) of the MR damper is not taken into consideration. Lee et al. [32] proposed methods for optimizing the magnetic field range of the MR damper and, also, the dynamic behavior. However, the proposed MR valve exhibited a response time between 28–125 ms, which is too long for use in semi-active car suspension. To secure effective control of the system, the response time of the MR damper should be smaller than approximately 1/10 of the highest controlled frequency of the system. If the response time is longer, the system is not able to react on the change of the damper state and the efficiency is lower [33,34]. Koo et al. [35] measured the transient response of the LORD Motion Master automotive damper. The overall response time of the damper was identified in the range of 15–55 ms. Based on the previous study, Strecker et al. [36] defined three main sources which prolong the response time: (i) MRF, (ii) electrical circuit—especially the coil inductance, and (iii) magnetic circuit. Subsequently, they designed a special current controller, minimizing the response time of the electrical circuit when switching on and also switching off [37] with the overvoltage method. The maximum measured response time caused by the electrical circuit was around 0.5 ms. Goncalves et al. [38] defined the response time of the MRF between 0.45 and 0.6 ms under varying magnetic field strengths at high shear rates. Jolly et al. [39] presented that the chain dynamics response time (rheological response time) of MRF depends on the viscosity of the currier fluid, particle concentration, and magnitude of the applied magnetic field. However, the main source of the overall delay of the damper is caused by the magnetic circuit (around 20 ms for the fastest valve), specifically by occurring eddy currents as a reaction on the extremely fast change in the magnetic field size. The eddy currents are generated in the magnetic circuit, which create the magnetic field acting in the opposite direction to that desired magnetic field [37]. The resulting field is therefore significantly lower. The size of the eddy currents is proportional to the rate of change in the generated magnetic field [40]. The higher the control speed, the more pronounced is the effect of the eddy currents, which greatly extend the final response time [41]. The influence of the eddy currents on the overall magnetic field in the MR device was published in the paper by Maas and Güth [42]. They used a ferrite as a material for the magnetic circuit in their experimental MR clutch and measured the response time between 4.1 and 5.3 ms. 

The problem of eddy currents can be solved by two approaches: (a) by means of material, and (b) by means of shape. The material solution is easier but is a compromise between the static and transient efficiency of the magnetic circuit. Materials like ferrites, permalloys, or soft material composites (SMC) generally have lower permeability or a lower magnetic saturation limit compared to pure iron [43,44]. However, they have high electrical resistivity, thus preventing the formation of the eddy currents. Other problems with these materials are their low mechanical strength, low impact resistance, high cost, and in the most cases poor machinability [45]. On the other hand, the material approach is commonly used in electrical engineering for electromagnetic valves or high frequency transformer and inductor cores in switching power supplies [46,47].

The second approach for prevention of eddy current formation is the shape approach [41,44]. For instance, this approach is applied at low frequency transformers using isolated sheets. A MR valve with a magnetic circuit composed of isolated sheets was also published in [48,49]. However, this solution allows the elimination of eddy currents only to a certain extent. For more significant elimination of the eddy currents as well as circuits of complicated shapes, the application of simple sheets is not suitable and it requires the use of complex shaped grooves intersecting the path of the eddy currents [50,51]. This way, the same response time as in the case of the material approach can be achieved. However, the complicated shape of the grooves is practically unmanufacturable by conventional methods because the grooves must also be structured within the core material. For the fabrication of such complex cores, additive technologies can be used, particularly the selective laser melting (SLM) method that allows 3D printing from metal powder [52]. 

This paper introduces a novel approach to the design of an ultra-fast MR valve based on materials with high electrical resistivity (material approach) or materials with low electrical resistivity (shape approach). This paper is a follow-up to earlier research made by the author’s team, where an ultra-fast MR damper with a magnetic circuit from ferrite was tested [37,53,54]. The low-impact resistance and bad machinability of ferrite prompted more study in the field of materials suitable for use in an ultra-fast MR valve. This paper includes an evaluation of individual approaches on the basis of the length of response time and dynamic force range. A transient simulation of the magnetic circuit was the main tool for material selection or groove designing. The output from the model is the time flow of the magnetic flux density when the electric current is switched on (step signal). From this course, the response time of the magnetic flux density can be determined. The simulation was carried out for six materials of the magnetic circuit with different electric conductivity (material approach) and for two grooved variants (shape approach) for achieving the shortest response time and high dynamic range. The simulated variants were verified by magnetic field measurement. Since the MR valve was purposely designed for use in an MR damper, the response time of the damping force was also measured. The knowledge from this study aims to contribute to the development of the new generation of ultra-fast magnetorheological elements, particularly semi-active controlled MR dampers.

## 2. Materials and Methods

The geometry of a simulated and tested MR valve is identical for all variants and is shown in the drawing in Figure 1. The coil has 120 turns.

### 2.1. Definition of Response Time

The simplest dynamic system which can serve as an approximation of dynamic behavior of an MR valve is a first-order system. The response of such a system on a step control signal is shown in Figure 2. The response is expressed by time constant *τ* (primary response time), which determines the time when 63.2% of maximal controlled value is achieved (see Figure 2). 

Because the behavior of the MR valve does not always correspond to the first-order dynamic system, the criterion of 90%—secondary response time *T_s_* of maximal controlled value—was also defined. This value is frequently used to determine the transient response of actuators in the industry. 

The rate between the secondary and primary response times indicates the level of correspondence between the measured system and its approximation by the first-order dynamic system. The rate between the secondary response time and primary response time for the ideal first-order dynamic system is 2.3 (see Figure 2).

Figure 3 shows the details of the force response time of the MR valve with the primary response time *T_p_* and secondary response time *T_s_*. Figure 3 is used simply for illustration. The data for this figure were obtained without fast current controller.

### 2.2. Transient Magnetic Simulation

A simulation model was created in Ansys Electronics Desktop 2016.1—Electromagnetic Suite 17.1. The output from the simulation is a response of the magnetic flux density on electric current (step signal) in the gap. 

The 3D model had to be used for simulation of grooved variants (shape approach), which could not be simulated by 2D symmetry. The axis (*y*-axis) and plane (x–z) symmetry were used within the model creation for reducing the calculation time (see Figure 4). Figure 4 also shows the constituent components of the model. All details such as groove for line of the coil power supply, fillets, or rounds of the corners or chamfer of edges were neglected for simplification of the mesh. All simulations for response time determination were carried out with MRF in the gap; however, for model verification by measurement of magnetic flux density, the air in the gap was prescribed because this measurement could not be realized with MRF. The eddy currents were switched off for faster calculation on electric non-conducting materials or materials with conductivity lower than 1 MS·m^−1^. This condition is met by MRF (0.01 MS·m^−1^), environment (vacuum), and the coil which, in the model, is represented by solid conducting material but, in fact, is combined from individual isolated wires. The details of the parts used in the model are listed in Table 1. The outer cylinder and core are fabricated from tested materials, and their properties are mentioned below in a separate table as well as *B-H* curves (Figure 5) for inputs to the model.

Boundary conditions were set as follows: (a) electrical insulation between the coil and core and between the coil and piston rod; (b) normal magnetic flux passes through the plane of symmetry (plane x–z); (c) coil as “stranded” with 60 turns (1/2 of total turns). The excitation was an electric current step with a rising edge duration of 0.3 ms. The resulting achieved magnetic flux density B was determined as an average value from the whole volume of the active zone (gap). This is especially important for results of grooved variants because the magnetic flux density is slightly different for the area in front of the groove and jag.

The finite element mesh was set differently for both the material and the shape approach. The mesh for the material approach was generated as a default mesh with subsequent fining leading to the resulting size, as shown in Table 2. Further fining of the mesh had negligible impact on the result. The final mesh setting of the shape approach is also stated there and described in detail below.

For the shape approach, the setting of the mesh was more complicated because the thin grooves intersecting the lines of eddy currents have to be modeled. The problem was identified in the model with a high number of thin grooves (0.1 and 0.3 mm thick). On the boundary of the grooves and solid body, the observed elements were too large. This boundary is the most important for the propagation of eddy currents and, therefore, the right set of mesh has a significant influence on the results. This problem was only related to the outer cylinder and core. Finally, the mesh was optimized so that the grooves had elements with a length of only 0.8 mm, and the rest of the part had a length of 1.35 mm. The smaller size of the elements was tested but without any significant change in results. The timestep of the solver was set to 0.1 ms.

### 2.3. Materials for the Magnetic Circuit

The materials listed in Table 3 were selected for testing, and their corresponding *B-H* curves are plotted in Figure 6. For pure iron, the *B-H* curve of Behanit (ČSN 19 991) was applied because this steel has only 0.01% carbon. Instead of the stainless steel AISI 420A *B-H* curve, the available curve of AISI 430, which has a little lower content of carbon, was used. *B-H* curve of Vacoflux 50 corresponds to the *B-H* curve of HiperCo 50, which should be the identical material offered under another trade name. The *B-H* curve is obtained from the website [55].

Cutting steel 11SMn30 was selected as the default material representing common magnetic circuits. This steel has a low carbon content, which leads to low remanence and high magnetic saturation. The steel composition ensures very good machinability and sufficient strength. 

Ferrite is mentioned in this paper because it was the first material that was tested as a material for an ultra-fast MR valve. Ferrite has high electrical resistivity but low magnetic saturation, which is limiting and causes low dynamic force range because the coil cannot generate a higher magnetic field in this material. However, ferrite has other and much worse properties—low resistance against impacts or shock loading, significant decrease of magnetic saturation with increasing temperature (25% decrease for a temperature rise from 25 to 100 °C), and bad machinability because of overly high hardness. Grinding with diamond tools is the only suitable process. Therefore, other materials were considered as a replacement for the ferrite.

Sintex SMC prototyping shows relatively high electrical resistivity and very good machinability, which ensures manufacturing by lathe-turning, milling, or drilling. On the other hand, the low permeability leads to high energy consumption when the magnetic field is generated. 

Vacoflux was selected as a high-tech alloy with a high magnetic saturation level and, also, relatively high resistivity resulting in low eddy-current generation. The disadvantage of this alloy for real applications is its very high price.

Pure iron was selected as the material for the fabrication of the magnetic circuit by SLM as a second representative of the shape approach. The SLM method enables 3D printing from metal powder. Pure iron has only trace concentrations of carbon and, therefore, it has low remanence, high permeability, and high saturation. After the tuning of the process parameters of the print, the ultimate strength was measured at around 460 MPa, which is an almost identical value to the cutting steel 11SMn30 [56]. 

Figure 7 compares the electrical current density 0.4 ms after turning the electric current on for core and outer cylinder made of cutting steel material with high electrical conductivity (b) with the core and outer cylinder made of Sintex material with low electrical conductivity (c). The electrical current density in the coil is the same for both cases, but it can be seen that the electrical current density (eddy currents) is not present in the core and outer cylinder made of Sintex material.

### 2.4. Shape Approach—Preventing the Eddy Currents by Shaping the Grooves

The shape approach is applied only to materials which have low electrical resistivity and where a response time would be long without the grooves. The orientation of the plane, which includes the dominant dimensions of groove (length and width), should be identical to the plane in which the closed loop of the magnetic line lies (see Figure 7a). Furthermore, the grooves should be placed over the areas with maximal magnetic flux density so that the contribution of the grooves should also be maximal. For simple geometry of the magnetic circuit, the course of magnetic flux lines can be determined using Ampere’s law (rule of the right hand); for complex geometry, it is better to use magnetostatic analysis of the solid steel variant (not grooved) of the magnetic circuit (Figure 8a). The groove oriented according to the displayed plane in Figure 7a intersects the flow of the eddy currents in Figure 8b, which prevents the generation of eddy currents in such an extension. However, the number and shape of grooves was proposed based on transient analysis to obtain a resulting response time because its accurate expression by analytical equations is impossible due to the result being influenced by several simultaneously acting phenomena: (i) the electrical resistance of the overall eddy current path in the magnetic circuit increases with the number and size of grooves—the path is longer (see Figure 8b); (ii) the length and direction of the path for eddy currents is influenced by the local change in conductivity of the magnetic circuit and by direction of the magnetic flux; (iii) direction of the magnetic flux is influenced by a local change in magnetic resistivity caused by the presence of the grooves; (iv) the magnitude of the magnetic flux is influenced by local magnetic saturation of the material around the grooves. However, in a grooved variant, the main contribution for achieving the shorter response time can be attributed to an increase in the total electric resistance caused by the flow of the electric current along the longer path and through places with a much smaller cross-section (see Figure 8). 

Since induced electric voltage based on Faraday’s law remains the same at grooved and full variants due to identical supply to the coil, the eddy currents must be reduced due to the increasing resistance. It is important to note that the grooves will almost not decrease magnetic conductivity because they do not intersect the magnetic flux lines and almost do not decrease the effective cross-section of the circuit in the direction of the magnetic flux lines. Only some local changes around the grooves will occur. The exact quantification of the number and shape of grooves on the response time of the magnetic flux density in the gap cannot be calculated by any analytical equations. Therefore, the magnetic 3D FEM model must be used.

In simulation, the influence of a grooved core and outer cylinder was discovered separately, due to the reduction in calculation time. If any groove shape leads to a decrease in response time with a grooved core or outer cylinder, the same trend of response time reduction can be observed when this shape is applied on both grooved parts. The final variant was selected from combinations of groove shape and number, including groove thickness in the range from 0.1 to 0.7 mm, groove depth from 1 to 8 mm, and groove number from 4 to 76. The fabrication limitations of the SLM method and electro-erosive manufacturing were also taken into consideration. The final design of the grooves is shown in Figure 9 and Figure 10. Both parts have 48 grooves. The groove thickness is limited by the wire thickness of electro-erosive manufacturing (0.35 mm) and by the process of vacuum casting of plastic into the grooves in order to inhibit the MRF flow through these grooves, which would lead to an increase in the cross-section of the gap and, consequently, a decrease in damping force. Grooved parts fabricated by 3D printing from iron powder had worse machinability due to the total absence of carbon, and the grooves were filled with removed material when lathe-turning. Therefore, the grooves on these parts were not cast by the plastic.

### 2.5. Measurement of the Magnetic Flux Density—Transient and Static Measurement

This measurement was carried out for verification of the simulation model. The output from the model was a time flow of magnetic flux density when the magnetic field is switched on or switched off. The simulations were calculated with the MRF in the gap. However, for comparison of the model and the measurement, the air instead of the MRF in the gap was set up because the measurement with MRF cannot provide accurate results. The thickness of the Hall probe is 0.55 mm, and the thickness of the gap is 0.65 mm. Therefore, the Hall probe takes almost all of the place between the magnetic poles in the gap, and since the relative permeability of the probe is 1, the acquired data would be drawn significantly by the presence of the probe (the permeability of MRF is from 4 to 6). However, if the model is valid for the air in the gap, the simulation with MRF should also be valid. Two types of measurement with almost identical configuration were carried out: (a) measurement of static properties of the magnetic circuit (dependence of magnetic flux density on input current to the coil); and (b) response of the magnetic flux density on the current step (see Figure 11). For response measurement, a patented current controller developed by our team was used instead of the common power supply. The controller drives the rise and drop of the current using the overvoltage technique described in [36,37]. The controller is supplied by a voltage of 30 V and the input signal is generated from a control computer. The current rise from 0 A to 2 A took between 0.2–0.6 ms depending on the variant of the piston, drop from 2 A to 0 A took up to 0.2 ms for all the variants. The magnetic flux density was measured using a teslameter (F.W. Bell 5180) with an ultra-thin transverse probe (STB1X-0201, F.W. Bell, Portland, OR, USA). The magnitude of electric current in the coil is obtained by measuring with a Fluke i30 current clamp. These two signals together with the voltage on the coil contacts are recorded and conditioned with a sampling frequency of 200 kHz by a front-end Dewetron USB-50-USB2-8 connected to a laptop.

### 2.6. Measurement of the Damping Force—Transient and F-v Curves Measurement

The tested MR valve was purposely designed to work in a hydraulic tube and can generate a damping force for the evaluation of the overall response time of an MR device, including the response of the electric circuit, magnetic field, and MR fluid. There are some simplifications (see Figure 12), such as low friction sealing and guiding (small leakage can occur), steep F-v curve (without bypass of the MR valve for modeling of the F-v curve), statically overdetermined system of piston guiding (sliding bearing and two bronze piston cups), or no sealing of the coil against MRF (only short-term testing considered).

For the measurements, a hydraulic damper tester by Inova was used (see Figure 13b). The damping force was measured by load gauge HBM U2AD1/2, the position of the piston rod by resistance sensor VLP15$A150, and the current was calculated from a voltage drop on a 0.1 Ω power shunt. These three signals were recorded and conditioned with a sampling frequency of 50 kHz by analyzer Dewe-800 by Dewetron (see Figure 13a). The piston velocity was calculated by derivation of the piston position. The temperature was monitored in order to not exceed 40 °C. If it did so, the test was temporarily interrupted.

Firstly, the *F-v–I* curves (dependence of damping force on piston velocity for different electric currents) were measured. The dynamic range of the valve can be determined from this dependency. The start-up test was applied with a frequency from 0.05 to 3 Hz, corresponding to a piston velocity of 0.01 to 0.45 m·s^−1^ with an amplitude of 24 mm. The *F-v*–*I* curves were measured for supplying currents of 0.0, 0.1, 0.2, 0.4, 0.5, 1.0, 2.0, and 2.5 A. The tester piston performed a harmonic motion. In post-processing analysis, only the damping force at the center of the stroke was selected. In this position, the velocity reaches maximum, and the acceleration is zero. 

For the transient measurement, the same current controller and hydraulic piston controller were used for the generation of the step signal as a measurement of magnetic flux density (see Figure 13). The piston movement was excited by a triangle signal of position, ensuring the unchanging velocity during the whole stroke. The measurement of the transient response was carried out for piston velocities 0.1, 0.2, and 0.3 m·s^−1^ and for the current in the range from 0 to 2.5 A, with an increments of 0.5 A. The current was switched on and switched off alternately always after two strokes. The exact moment of switching on or off was in the center of the stroke (see Figure 3 and Figure 14).

## 3. Results and Discussion

### 3.1. Transient Responses of Magnetic Flux Density in the Gap with Air

The material approach was primarily considered as a basis for the reduction of the response time. Figure 15 indicates the achieved response time dependent on representative electrical resistivity. The results are obtained as an output from the transient simulation and are valid only for the specific geometry of the MR valve shown in Figure 1, and the current rise and drop in the coil within 0.3 ms, which corresponds to the used current controller. Although the absolute value of the response time is valid only for this specific case, the trend is valid generally. This analysis can be useful particularly for designers of semi-active MR devices. For instance, if a response time shorter than 2 ms is needed (marked in blue in Figure 15), the materials lying under the line must be used. However, if a shape approach on materials above the line is applied, the required response time can also be achieved. The higher the material above the line, the more complex the solution of the shape approach (grooves). For instance, it means that only several grooves for stainless steel AISI 420A are needed, contrary to pure iron. On the other hand, the application of a material with higher electrical resistivity than 10^−4^ Ω∙m seems superfluous.

For the verification of transient simulation, six materials were selected (see Table 3). When the response time was longer than required, the shape approach was applied. All simulated and tested variants with the achieved results of magnetic flux density measurement are mentioned below in Table 4. The number of grooves for Vacoflux 50, pure iron from SLM, and 11SMn30 was proposed on the basis of transient simulation to achieve the required response time of 2 ms or better in a model with MRF. It has to be noted that the following results are measured and simulated with air in the gap. Hence, the resulting response time is lower with air than with MRF. This is caused by the approximately four times lower relative permeability of the air. Since the gap displays the major magnetic resistance of the circuit (typically more than 90%), the MRF presence leads to a much higher magnetic flux and corresponding magnetic field intensity, which causes a more significant influence of the occurring eddy currents and a longer rise of magnetic field at the identical speed, because a much higher value has to be achieved. For material AISI 420A, more grooved variants were suggested in order that the contribution of a higher number of grooves was illustrated.

The measurement and simulation resulted in following conclusions:MR valves made of ferrite materials exhibit the fastest response time. Ferrite material also exhibits the lowest remanence; however, only a small magnetic flux density can be achieved.Valves made of Sintex (SMC) are as fast as those of ferrite, despite a lower electrical resistivity than the ferrite. A magnetic circuit made of Sintex material achieves just as short a response time without grooves. The magnetic flux is significantly higher than in the case of ferrite.Vacoflux variants exhibit very high magnetic flux density and lower response time of magnetic flux density than any metal material. Six grooves reduced the response time almost to the level of the Sintex valve.The SLM sample (with and w/o the grooves) shows a high magnetic flux density because of the high permeability/magnetic saturation limit of the pure iron powder. The response time of the valve without grooves is the longest of all measured samples. This result was expected due to the lowest resistivity of pure iron. On the contrary, the version with grooves showed a more than 10 times shorter response time.The AISI 420A valve without grooves shows quite a low response time and high magnetic density, but also the highest remanence. High remanence is reducing the dynamic force range of the valve and it is also not advantageous for a fast current controller.In the 11SMn30 sample, an identical number of grooves as for the SLM must be fabricated in order to achieve a similar response time as the Sintex valve. The magnetic flux density is higher than in case of the Sintex valve, but slightly lower than the valve made of pure iron by SLM. During e fabrication, the parts of the magnetic circuit were cooled and the cutting speed was low in order to prevent overheating. Thanks to this, the valve exhibits low remanence and high permeability.There is some difference between the simulated and measured values. This is probably caused by input *B-H* curves and electrical resistivity of materials being taken from datasheets. Previous research has shown that the exact composition of alloys and, thus, *B-H* curves can differ for a smelting batch. In addition, the heat treatment and mechanical loading can change the magnetic and electrical properties. For more precise results, *B-H* curves and the resistivity of materials used for the circuit must be measured individually. However, the presented model is an effective tool for determining the transient behavior of a magnetic circuit.

### 3.2. Transient Responses of MR Damper Force

Figure 16, Figure 17, Figure 18, Figure 19, Figure 20, Figure 21 and Figure 22 show the detail of rises and drops of force in a response in the electric current step. The orange dashed line shows the force level corresponding to primary response *τ*_63_, *τ*_36_ time and the green dashed line shows the level corresponding to secondary response time *τ*_90_, *τ*_10_. The overall results based on the average from three measurements are summed up in Table 5. The table also compares the overall time response of the MR valve with the response time of the simulated magnetic field as a most important source of the overall MR valve response time. The influence of MR fluid and compliance of the hydraulic system is not incorporated in the simulation results.

The measurements showed the following conclusions:The response times of MR valves made of solid parts corresponds with their electrical resistivity.Grooves rapidly speed up the response time of MR valves made of metals.The force response course is similar to first-order dynamic systems only for pistons made of solid steel. In this case, the secondary response time is approximately 3 times longer than the primary response time.The secondary response time for pistons made of materials with high electrical resistivity is much shorter. The response of the force is copying the course of electric current with some delay. Approximation by a first-order dynamic system is therefore not very suitable.The simulations for grooved pistons or pistons made of materials with high electrical resistivity predicted significantly shorter response time of the magnetic field (basically the same as the response time of the current). The difference between the measured and simulated values can be explained by the response time of the MR fluid itself. There is always a time delay of approximately 0.6 ms between the beginning of the electric current rise (or drop) and the force rise (or drop). This phenomenon is practically not dependent on piston material, or velocity.The best dynamic range can be achieved with the piston made of Hiperco/Vacoflux.The course of force in the case of pistons with a force secondary response time up to 3 ms exhibits quite heavy oscillations. This is probably caused by the compliance of MR fluid and other parts of the MR damper. This problem was explained in [57].

## 4. Conclusions

The response time of a MR damper can be reduced by the reduction of eddy currents during fast changes of the magnetic field in an electrical current. The eddy currents can almost be eliminated by the use of special materials with very high electrical resistivity such as ferrite or soft magnetic composites. The use of these materials, however, significantly reduces the dynamic range due to the lower permeability or lower magnetic saturation level. In addition, the mechanical properties are poor. However, total eddy current elimination is not necessary because of the response time of the magnetorheological fluid itself. Efforts to reduce the response time of the magnetic flux density in the gap to shorter values than the response time of the MR fluid cannot bring a significant reduction of the damper’s force response time. From this point of view, the shape attitude can bring advantages. If the eddy currents in steel are reduced by grooves so that the response time of the magnetic flux density will be in the same range as the response time of the MR fluid, an overall response time comparable to that of the ferrite or SMC magnetic circuit can be achieved in the MR damper. In addition, the dynamic range and mechanical properties can be significantly better due to the better magnetic and mechanical properties of steel. The measurements also showed some phenomena which are, in most of cases, not reflected in simulations employing MR devices. The waveform of the force in a response in the control signal step differs for different constructions of MR valve. Whereas the response time of the MR valves made of iron, steel, or Hiperco/Vacoflux without grooves can be considered as first-order dynamic systems, the secondary response time of MR valves made of materials with high resistance or grooved variants is much shorter. In other words, the force response in the control signal step does not correspond to the exponential function. In addition, there is always a delay between 0.6 and 0.8 ms between the control signal and the start of the force change, which is not significant for MR valves with a long response time; however, neglecting it in fast semi-active system simulations can cause low fidelity of model behavior.

## Figures and Tables

**Figure 1 materials-14-02500-f001:**
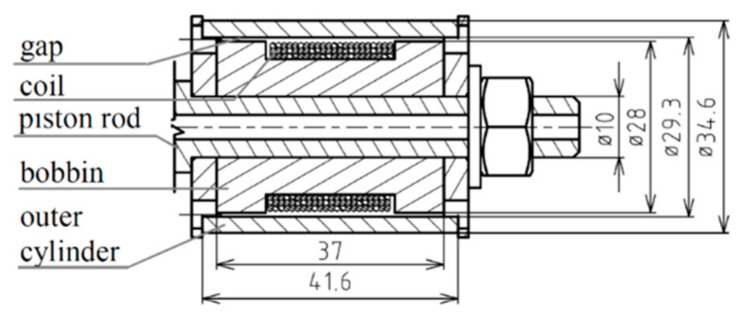
Geometry of the MR valve used for simulation and testing.

**Figure 2 materials-14-02500-f002:**
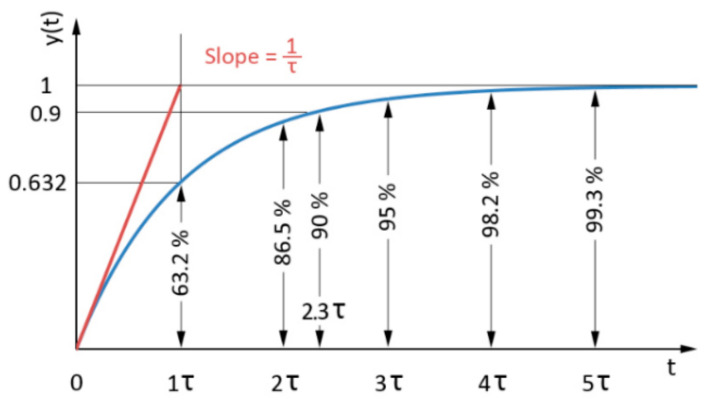
Time response description of the first-order dynamic system to a unit step.

**Figure 3 materials-14-02500-f003:**
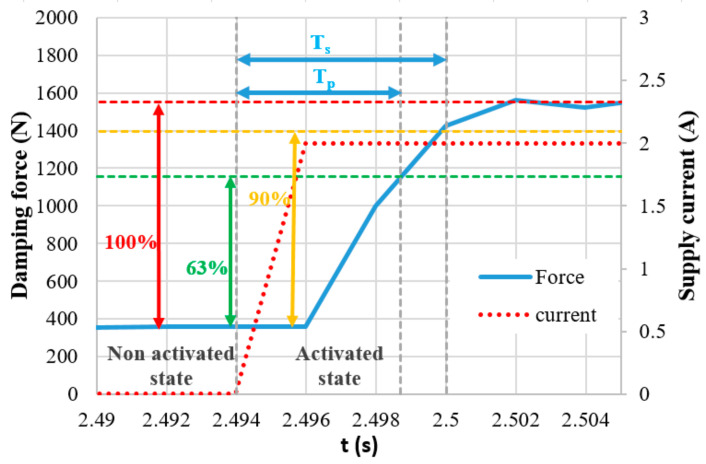
Detail of the damper’s force response in electric current step 0–2 A with determination of the primary response time *T_p_* and secondary response time *T_s_*.

**Figure 4 materials-14-02500-f004:**
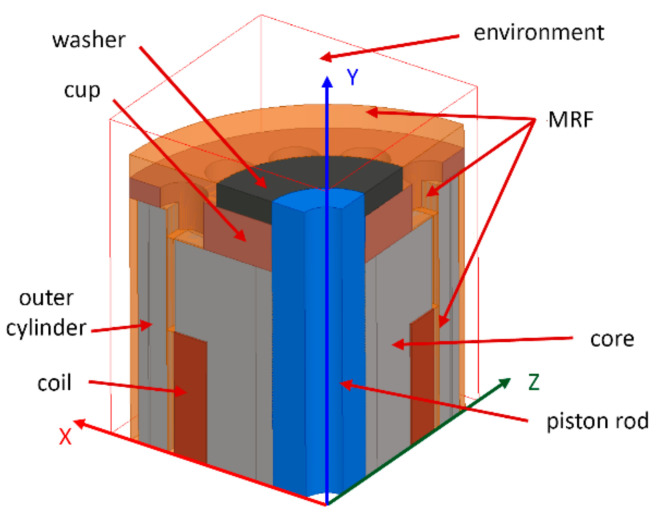
Axis and plane symmetry of the model.

**Figure 5 materials-14-02500-f005:**
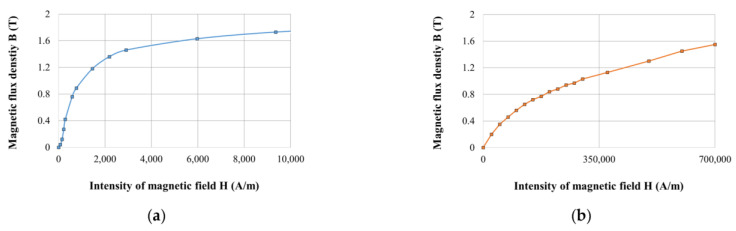
(**a**) *B-H* curve of steel used for piston rod (own measurement); (**b**) magnetorheological fluid LORD MRF-132DG (datasheet).

**Figure 6 materials-14-02500-f006:**
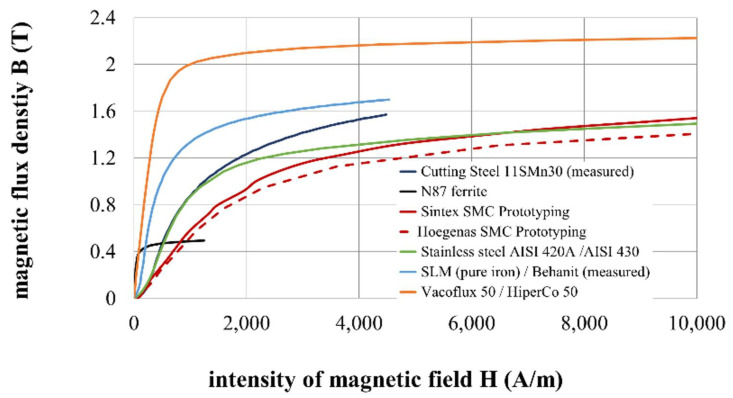
*B-H* curves of materials used in the simulation model.

**Figure 7 materials-14-02500-f007:**
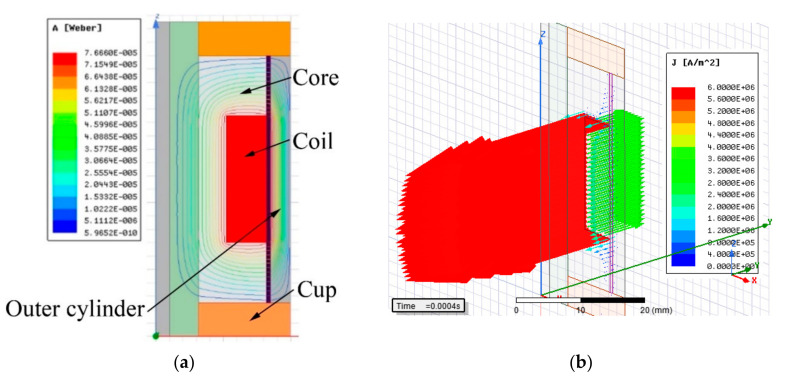
Explanatory example of eddy current generation (**a**) magnetic flux lines in a variant with solid steel core and outer cylinder; (**b**) eddy currents in the variant with solid steel core and outer cylinder; (**c**) eddy currents in a variant with Sintex core and outer cylinder (for comparison) at the identical time and exciting current—eddy currents are generated only in a piston rod which is made from structural steel; however, the rate of eddy currents is much lower because of the much lower magnetic field in this location.

**Figure 8 materials-14-02500-f008:**
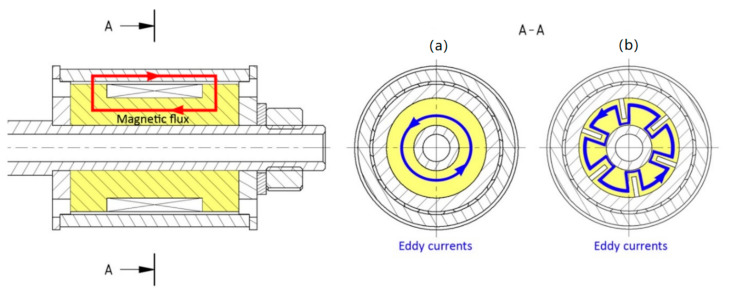
The comparison of eddy current paths in the core (**a**) without grooves; and (**b**) with grooves.

**Figure 9 materials-14-02500-f009:**
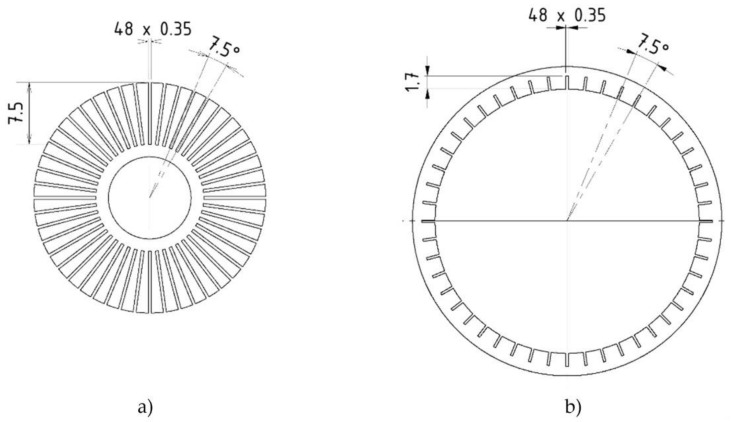
Drawing of grooved (**a**) core; and (**b**) outer cylinder.

**Figure 10 materials-14-02500-f010:**
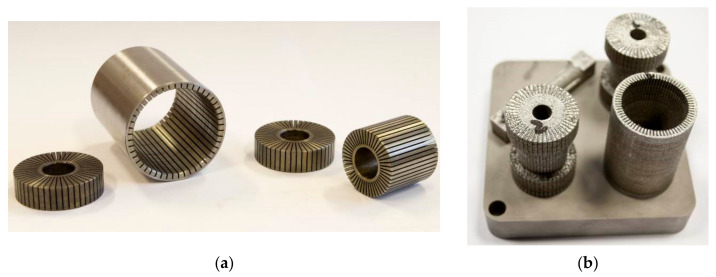
Outer cylinder and parts of core fabricated by (**a**) electro-erosive manufacturing; (**b**) SLM—3D printing.

**Figure 11 materials-14-02500-f011:**
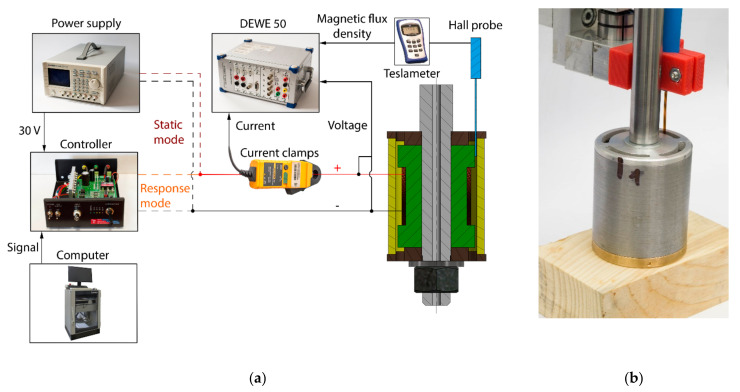
(**a**) Scheme of measurement configuration; (**b**) detail from measurement of MR valve with Hall probe.

**Figure 12 materials-14-02500-f012:**
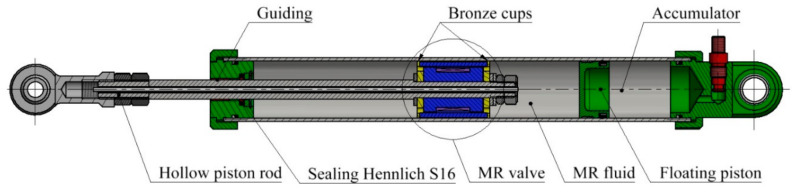
Testing device for measurement of overall response time of damping force.

**Figure 13 materials-14-02500-f013:**
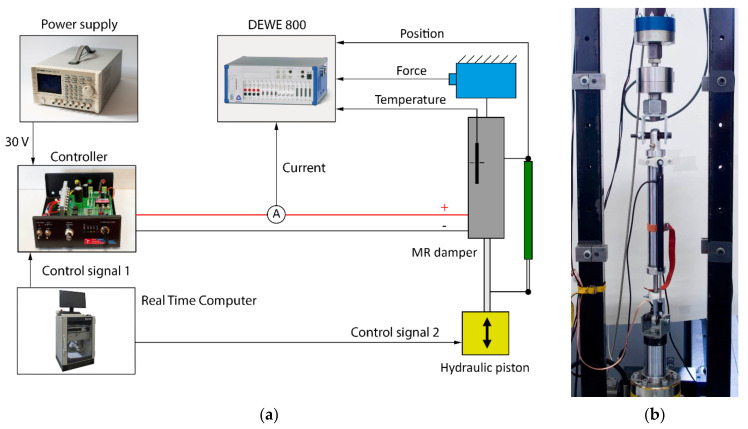
Measurement of damping force on tester Inova: (**a**) scheme; (**b**) tester with MR device.

**Figure 14 materials-14-02500-f014:**
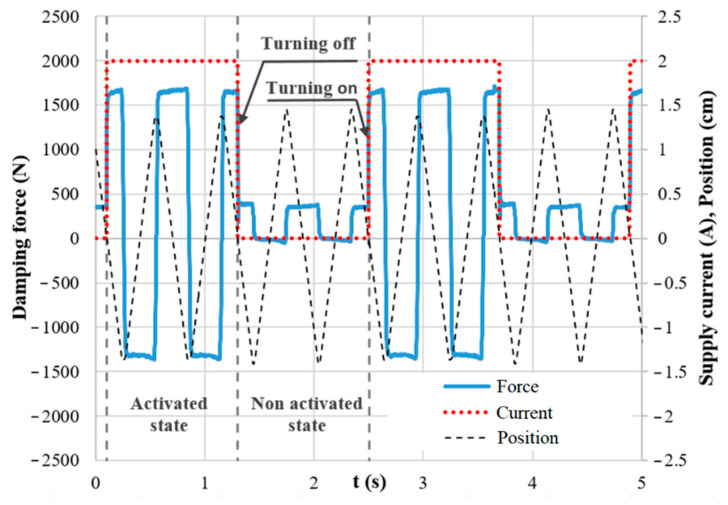
The force courses, input control, and position signal during response time measurement.

**Figure 15 materials-14-02500-f015:**
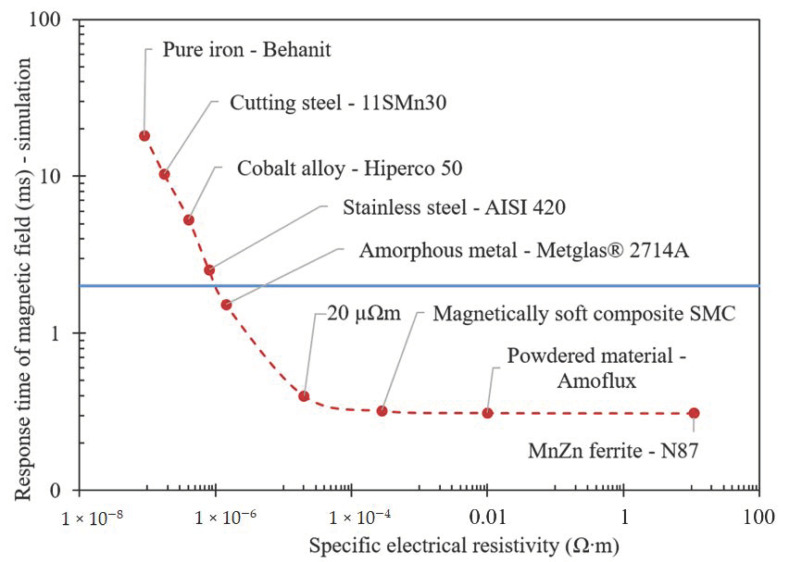
Material approach—response time of magnetic flux on step signal of 2 A for materials with various electrical resistivities (valid for specific geometry of MR valve with MRF).

**Figure 16 materials-14-02500-f016:**
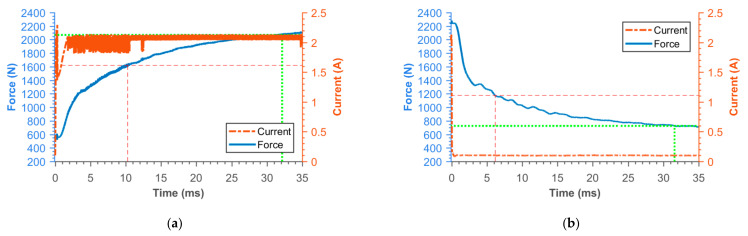
Detail of force rise (**a**) and drop (**b**) of 11SMn30 solid piston.

**Figure 17 materials-14-02500-f017:**
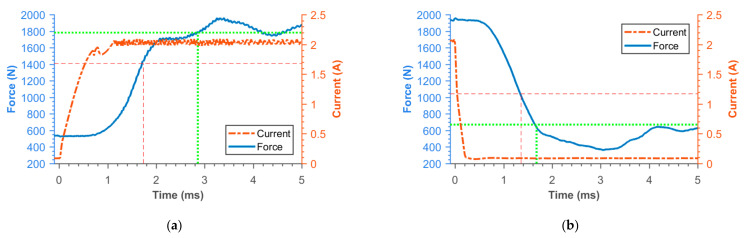
Detail of force rise (**a**) and drop (**b**) of 11SMn30 grooved piston.

**Figure 18 materials-14-02500-f018:**
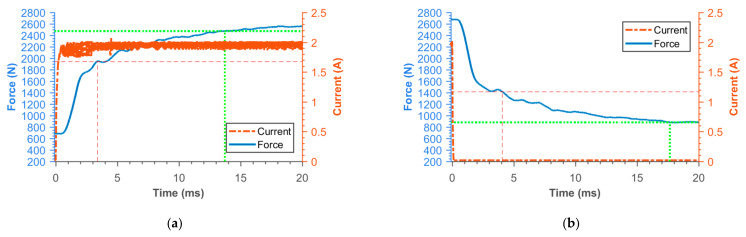
Detail of force rise (**a**) and drop (**b**) of 11SMn30 Hiperco/Vacoflux piston.

**Figure 19 materials-14-02500-f019:**
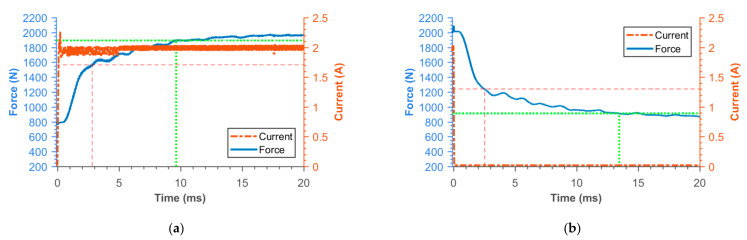
Detail of force rise (**a**) and drop (**b**) of AISI420 piston.

**Figure 20 materials-14-02500-f020:**
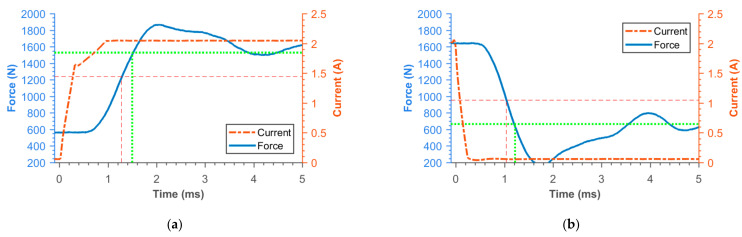
Detail of force rise (**a**) and drop (**b**) of SMC piston.

**Figure 21 materials-14-02500-f021:**
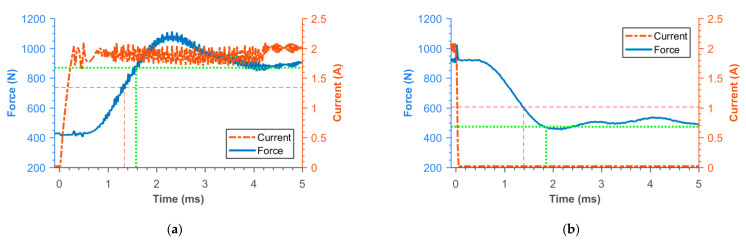
Detail of force rise (**a**) and drop (**b**) of piston consisting from Ferrite N87 bobbin and 11SMn30 outer cylinder.

**Figure 22 materials-14-02500-f022:**
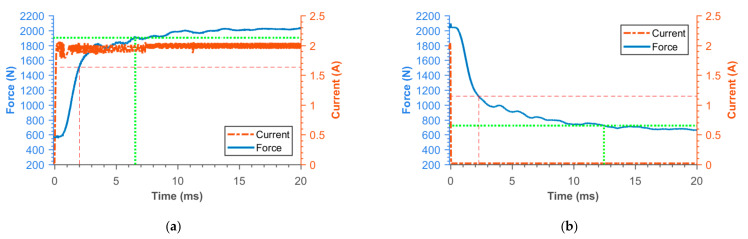
Detail of force rise (**a**) and drop (**b**) of SLM (pure iron) grooved piston.

**Table 1 materials-14-02500-t001:** Description of materials used in the model.

Part	Material	Electrical Conductivity (MSm^−1^)	Electrical Resistivity (10^−6^ Ω∙m)	Eddy Currents	Relative Permeability
Piston rod	S235JRG	6.3	0.16	Yes	*B-H* curve
Washer	S235JRG	6.3	0.16	Yes	*B-H* curve
Coil	copper	58.0	0.02	No	1
Cup	bronze	10.0	0.10	Yes	1
Environment	vacuum	0	-	No	1
MRF	MRF-132DG	0.01	100	No	*B-H* curve

**Table 2 materials-14-02500-t002:** The RMS edge length of mesh elements.

Size of Element (mm)	Piston Rod	Coil	Cup	Environment	MRF	Outer Cylinder	Core
Material approach	0.9–1.8	1.6	1.4–1.8	2.2–2.7	0.8–1.1	0.9–1.3	1.5
Shape approach	0.9	1.2	1.3	1.0	0.7	0.9	max 1.35 and 0.8

**Table 3 materials-14-02500-t003:** List of selected materials.

Material	Ultimate Strength (MPa)	Electrical Resistivity (10^−6^ Ω∙m)	Magnetic Saturation (T)	Relative Permeability	Machinability	Price
Cutting steel 11SMn30	470	0.17	1.9	1200	very bad	Low
N87 ferrite	30	10,000,000	0.5	6400	very bad	Medium
Sintex SMC prototyping	75	2800	1.45	430	good	High
Stainless steel AISI 420A	760	0.60	1.6	950	good	Low
SLM (pure iron)	460	0.13	1.7	1900	good	High
Vacoflux 50	350	0.42	2.35	3850	good	High

**Table 4 materials-14-02500-t004:** Overview of tested variants using material and shape approach and results of measurement of the magnetic flux density (with air in the gap).

Material	Measurement					Model	
	Grooves No. in Core/Outer Cylinder (-)	Max. Magnetic Flux Density at 2 A (mT)	Remanence at 0 A (mT)	Response Time Signal Rise (ms)	Response Time Signal Drop (ms)	Response Time Signal Drop (ms)	Max. Magnetic Flux at 2 A (mT)
11SMn30	0	173	14	1.43	1.61	2.1	202
11SMn 30	48/48	187	15	0.35	0.35	0.52	200
Hiperco/Vacoflux 50	0	208	8	1	0.92	0.68	223
Hiperco/Vacoflux 50	3/0	207	8.2	0.72	0.69	0.40	223
Hiperco/Vacoflux 50	6/0	206	6.6	0.67	0.61	0.32	223
stainless steel AISI 420A	0	180	33	0.93	0.68	0.67	211
SLM (pure iron)	0	230	16	2.21	2.41	3.62	215
SLM (pure iron)	48/48	221	16	0.49	0.45	0.35	217
Ferrite N87	0	134	3	0.31	0.35	0.23	156
Sintex SMC	0	152	11	0.35	0.29	0.27	184
Sintex (core) 11SMn30 (cylinder)	0/48	190	14	0.64	0.61	0.57	197

**Table 5 materials-14-02500-t005:** Performance comparison of pistons.

Type of Piston	Measurement			Model		
	Primary Force Response Time *τ*_63_ (ms)	Secondary Force Response Time *τ*_63_ (ms)	Dynamic Force Range (-)	Primary Response Time of Magnetic Field (ms)	Secondary Response Time of Magnetic Field (ms)	Flux Density at 2 A (mT)
Solid piston (11SMn30)	10.3	32.7	5.73	7.6	21.9	505
Groved piston (11SMn 30)	1.74	2.86	5.05	0.61	3	481
Hiperco/Vacoflux 50 piston	3.57	14.14	6.4	4.45	13.5	689
AISI 420A piston	2.82	9.69	3.17	2.38	6.2	452
SLM (pure iron) 48/48	2.03	6.78	5.25	0.78	4.3	516
Ferrite N87 bobbin + 11SMn30 outer cylinder	1.34	1.54	3.5	0.39	3.05	361
SMC piston	1.28	1.49	4.03	0.23	0.38	467

## Data Availability

The data presented in this study are available on request from the corresponding author.

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
