# Peer review of "Novel Approaches to the Design of an Ultra-Fast Magnetorheological Valve for Semi-Active Control"

_materials, 2021, doi:10.3390/ma14102500_

Round 1

Reviewer 1 Report

This contribution deals with the design of an ultra-fast magnetorheological valve for semi-active applications. By the literature research it turned out that the eddy currents are manly decisive for occurring time delay. The dynamic behavior and specific values are extensively introduced. Besides, certain materials of different shapes are investigated regarding this issue. The main focus lays on radially groves in the flux guiding parts to increase the electrical resistivity and thus are able to reduce eddy currents. Experimental investigations of the dynamic behavior of the magnetic flux comprehend the impact of different materials and shapes. In addition, experimental investigations of the force’s dynamic within an MRF-damper prove that grooves can rapidly speed up the response time of MR valves.

In my opinion the paper is very well written and can be recommended for publication. Before publication some points of improvement and extensions should be considered:

  • The idea of applying grooves in magnetic circuits is already published in US10480674B2 and commonly used approach in designing magnetic circuits to improve their transient response. Consequently, it has to be considered as state of the art.
  • The idea of applying grooves in MRF-valves of dampers was already introduced in https://doi.org/10.1016/j.ymssp.2019.02.058. Consequently, it has to be considered as state of the art.
  • Some sections include to many details which are not important for the reader of a journal paper. My suggestion is to reduce the manuscript by 3 pages. Especially, section 2 should be shortened for example the materials in section 2.3 or the description of the test setup. A further example, Fig. 2 can be expressed by an simple equation and is in general contained in Figure 3.

Further points to be improved:

  • Why is the size of the FEA grid in the gap (MRF) varying? I suggest to apply the smallest value of 0.7 for all studies for a better comparison.
  • I think the reference to Fig 7 is incorrect, currently Fig 8 is named in the text.
  • Figure 8: a) legend of the materials is missing; b) & c): Legend is to small to be read, is green the coil current? Please name it in the picture.
  • The content of figure 7 and figure 9 is very similar, maybe you remove figure 7 and use figure 9 instead.
  • Table 4: Please explain in the paper why the magnetic flux density of 11SMn30 with grooves is higher than without grooves.
  • Section 3.2 includes many reference errors.
  • In general from my point of view, all units at figure axis are not in accordance with ISO 80001. Please don’t use […] for units.
    Example: Instead of writing “t [s]” use “t in s” or even better “t/s” (normalized version).  

Reviewer 2 Report

OVERVIEW

The authors of the article deal with the design of an ultra-fast magnetorheological valve using two approaches for achieving a short response time. The authors of the article used FEM to find suitable materials and techniques as a result of a minimum response time of the magnetic flux density on the step signal of the current. The simulated results of the response time are compared to the measured results on the simple magnetorheological valve. A total of six different materials were tested. To prevent eddy currents in the examined samples, the authors of the article used a special technique involving grooves on materials with high electrical conductivity. The authors of the article found out that the best dynamic range can be achieved with the piston made of Hiperco/Vacoflux.

POSITIVE ASPECTS

1. Based on a literature review, the authors made quite an extensive overview of the issues related to magnetorheological fluids, especially for applications of material used as a damper or valve in cars.
2. In the introduction, the authors of the article clearly explained the problems related to eddy currents, long response time, and ways to limit the generation of eddy currents.
3. The authors of the article used adaptive technology enabling 3D printing from metal powder for the production of complex cores.
4. The authors of the article introduced a novel approach to the design of an ultra-fast magnetorheological valve based on materials with high electrical resistivity or materials with low electrical resistivity.
5. The authors of the article used a transient simulation of the magnetic circuit made of six different materials as the main tool for material selection or groove designing.
6. The authors of the article set a clear goal to contribute to a new generation of ultra-fast magnetorheological elements, especially semi-actively controlled magnetorheological dampers.
7. The authors of the article made a detailed description of the construction of a grooved core and outer cylinder.
8. Using a transient simulation, the authors found that the use of a material with a higher electrical resistivity than 10–4 Ωm for the magnetorheological valve is superfluous.
9. The authors of the article found out that the approximation by 1st order dynamic system is not very suitable for pistons made of materials with high electrical resistivity
10. The authors use the comparative method to describe the measurement and simulation results.

ISSUES

The presented work is useful but has some issues that need to be removed. I have a few comments that can be used to improve the article.

Minor issues
1. Signs of physical quantities need to write in italics according to ISO 31-4 (ISO 80000-5: 2007). Corrections need to be done throughout the article including a description of the image.
2. The authors of the article begin a description of the axes in graphic dependencies in capital letters. In Figure 5, the authors made an exception with the description of the axes begins in lower case. A uniform way of marking the axes in the graphs should be used.
3. Figure 5 lacks the designation of graphical dependencies as indicated in the description of the figure. Correct accordingly throughout the article.
4. Figure 8 lacks the designation of graphical dependency (a) as indicated in the description of the figure. Corrections need to be done.
5. The current rise and drop of the current controller 0.3 ms given in line 369 does not correspond to the value of the time increase in line 317. This fact must be justified in the text.
6. There is an extra space in the description of Figure 23. Corrections need to be done.

Major issues
1. The meaning of the acronyms “MR” and “SLM ” is not explained at all. Correct accordingly throughout the article.
2. The authors of the article refer in line 131 to an incorrect Figure 4, which shows the 3D geometry of the model. Corrections need to be done.
3. In the time course of the response to the step control signal in Figure 2, the timeline designation is missing. Corrections need to be done.
4. On line 135 and line 142, I see a mismatch between the designation secondary response time and primary response time. Corrections need to be done.
5. It is not clear to me why the authors of the article state the input signal in Figure 3 and Figure 16 and in the description of the vertical axis supply current. Are these two different quantities? An appropriate correction needs to be made.
6. According to Figure 12, the output from the Hall probe is connected directly to DEWE 50. In the text, the authors of the article state that the magnetic flux density is measured using a teslameter. A corresponding correction needs to be made in the text and the circuit diagram in Figure 12 so that there is no doubt about how to measure the magnetic flux density.
7. The authors of the article state that the piston speed was calculated by deriving the position of the piston. The time course of the velocity shown in Figure 14 does not correspond to the derivative of the time course of the piston position. An appropriate correction must be made so that the time course of the velocity corresponds to the derivative of the time course of the pistol position.
8. The authors of the article claim that: “The piston movement was excited by a triangle signal of position, ensuring the unchanging velocity during the whole stroke”, referring to Figure 15. However, Figure 15 does not show any time course. An appropriate correction needs to be made.
9. Line 356 incorrectly states the unit of speed. An appropriate correction needs to be made.
10. There are errors in some references in the text, for example, in lines 437, 441. Correct accordingly throughout the article.
11. The green dashed line used in Figures 18 to 24 is very thin and difficult to see. Use a thicker line.

REMARKS

1. I don't think the description of Figure 7 is complete. An appropriate correction needs to be made.

RECOMMENDATIONS

1. In the description of Figure 2, it is suitable to use more appropriate text describing the time response of the system to a unit step.
2. The units of physical quantities given in the description of the graph are usually given in round brackets. I recommend making a corresponding correction in the whole article.
3. I did not find any comment on the increased oscillations of the regulator in the case of the time course of the current in Figure 23, which results in oscillations in the time course of the increase in force. A brief comment on this fact needs to be added. 

QUESTIONS

I have two questions for the authors of the article.

1. The authors of the article state in line 173 that the rising edge duration of an electric current step is 0.3 ms. According to the time dependence of Figure 3, the time between 0 A and 2 A is equal to 2 ms. How do the authors explain this difference? The appropriate comment needs to be added to the text.
2. Does Figure 14 show the actual measured time course of the position? A brief comment on the time courses in Figure 14 needs to be added.

CONCLUSION

Regretfully, the paper cannot be accepted in its present form. Deficiencies need to be corrected.
